# Flux Increase Occurring When an Ultrafiltration Membrane Is Flipped from a Normal to an Inverted Position—Experiments and Theory

**DOI:** 10.3390/membranes12020129

**Published:** 2022-01-21

**Authors:** Ladan Zoka, Ying Siew Khoo, Woei Jye Lau, Takeshi Matsuura, Roberto Narbaitz, Ahmad Fauzi Ismail

**Affiliations:** 1Department of Civil Engineering, University of Ottawa, 161 Louis Pasteur, Ottawa, ON K1N 6N5, Canada; ladanzoka@yahoo.co.uk (L.Z.); narbaitz@uottawa.ca (R.N.); 2Advanced Membrane Technology Research Centre (AMTEC), Universiti Teknologi Malaysia, Johor Bahru 81310, Johor, Malaysia; yingsiew520@gmail.com (Y.S.K.); afauzi@utm.my (A.F.I.); 3Department of Chemical and Biological Engineering, University of Ottawa, 161 Louis Pasteur, Ottawa, ON K1N 6N5, Canada

**Keywords:** ultrafiltration, water flux, inverted position, theory, Bernoulli’s principle

## Abstract

The effects of flipping membranes with hydrophilic/hydrophobic asymmetry are well documented in the literature, but not much is known on the impact of flipping a membrane with dense/porous layer asymmetry. In this work, the pure water flux (PWF) of a commercial polyethersulfone (PES) membrane and a ceramic ultrafiltration (UF) membrane was measured in the normal and inverted positions. Our experimental results showed that the PWF was two orders of magnitude higher when the PES membrane was flipped to the inverted position, while the increase was only two times for the ceramic membrane. The filtration experiments were also carried out using solutions of bovine serum albumin and poly(vinylpyrrolidone). A mathematical model was further developed to explain the PWF increase in the inverted position based on the Bernoulli’s rule, considering a straight cylindrical pore of small radius connected to a pore of larger radius in series. It was found by simulation that a PWF increase was indeed possible when the solid ceramic membrane was flipped, maintaining its pore geometry. The flow from a layer with larger pore size to a layer with smaller pore size occurred in the backwashing of the fouled membrane and in forward and pressure-retarded osmosis when the membrane was used with its active layer facing the draw solution (AL-DS). Therefore, this work is of practical significance for the cases where the direction of the water flow is in the inverted position of the membrane.

## 1. Introduction

The effects of flipping membranes with hydrophilic/hydrophobic asymmetry are well documented in the literature. For example, Wang et al. [1] reported that flipping a porous sheet with one hydrophilic side and one hydrophobic side changed the tendency of water seeping into the pores, depending on whether the water droplets were placed on the hydrophilic or on the hydrophobic side. A Janus membrane with stacked hydrophobic and hydrophilic layers showed unidirectional water flow when the hydrophobic surface was exposed to a water-in-oil emulsion [2]. It is also known that in the pervaporation process, acetic acid/water selectivity is altered by flipping hydrophobic/hydrophilic composite membranes [3,4]. Interestingly, the effects of flipping a membrane with dense/porous layer asymmetry have not been investigated as much as those of flipping a membrane with hydrophilic/hydrophobic asymmetry.

The design of membranes with an asymmetric structure—either integrally skinned asymmetric membranes or thin-film composite (TFC) membranes—wherein a selective thin layer is supported by a porous sublayer to enhance mechanical strength, is currently the norm [5,6,7,8]. In the normal position, the dense skin layer is in contact with the feed solution, and the permeate leaves the membrane from the other side of the membrane in pressure-driven filtration processes such as reverse osmosis (RO), nanofiltration (NF), ultrafiltration (UF), and microfiltration (MF) [9,10]. Generally, membranes are not used in the inverted position, as the direct contact of their porous sublayer with the feed solution could lead to severe fouling. In some cases, however, the water flows from the large-pore-size surface to the smaller-pore-size surface, e.g., when the membrane is back-washed [11,12,13] or is used with the active layer facing the draw solution (AL-DS) mode in forward osmosis (FO) or pressure-retarded osmosis (PRO) [14,15,16].

It was reported as early as in 1970 in Sourirajan’s book *Reverse Osmosis* that the flux increased significantly when an asymmetric RO membrane was used in the inverted position as compared to the normal position [17]. When the feed pressure was increased, the flux increased as concave-up, but when the pressure was decreased from its maximum, the flux decreased linearly, resulting in a hysteresis curve. According to Sourirajan, this phenomenon was due to the partial pore expansion by the water flow caused by the trans-membrane pressure difference. Since Sourirajan’s early finding, this phenomenon has not been thoroughly investigated, probably because severe internal concentration polarization (ICP) and fouling are expected to occur in the porous layer when it faces the feed solution that contains electrolytes or organic contaminants. 

In our previous communication, we reinvestigated this phenomenon by conducting RO/NF experiments with two commercial polyamide TFC membranes in their inverted position [18]. It was confirmed that a significant flux increase occurred not only for the cellulose acetate asymmetric membrane, which was used by Sourirajan, but also for TFC membranes. However, unlike Sourirajan’s work, no hysteresis curve emerged when the pressure was changed from low to high and then returned to low values. Practically, no salt rejection was observed, and there was a severe flux decrease when a sodium alginate solution was filtered for 90 min. This was likely due to membrane compaction and fouling by the macromolecular solutes. It was also concluded by a model simulation that flux increase from the normal to the inverted position would not occur when the membrane was used in FO. 

The objective of this work, as the continuation of our previous study, was to know if a similar flux increase occurs when a UF membrane is used in the inverted position. For this purpose, (a) two flat sheet commercial polyethersulfone (PES) UF membranes and (b) two tubular ceramic membrane modules were tested in their normal and inverted positions. For both polymeric and ceramic membranes, filtration experiments were conducted to measure the pure water flux (PWF) and the flux of macromolecular solutions. As well, we attempted to explain the experimental observation by a model simulation based on the Bernoulli’s principle. This work is of practical significance for cases where the direction of the water flow is in the inverted position of the membrane, e.g., when backwashing of a fouled membrane occurs and in the FO/PRO process.

## 2. Materials and Methods

### 2.1. Materials

In this study, a commercial polyethersulfone (PES) MK UF membrane with a nominal molecular weight cutoff (MWCO) of 30 kDa supplied by Synder Filtration (Vacaville, CA, USA) was evaluated for its performance. A ceramic tubular UF membrane with a nominal MWCO of 100 kDa (T7/60/250) supplied by Shandong Jinhuimo Technology Co. Ltd. (Qingdao, China) was also used. The ceramic membrane composed of alumina and zirconia had seven feed channels, each of which had a 6 mm diameter and a length of 1.178 m, and its effective filtration area was 0.155 m^2^. Bovine serum albumin (BSA, MW: 66 kDa, Purity: 96%) was purchased from Sigma Aldrich (St. Louis, MO, USA). Both polyvinylpyrrolidone (PVPK90, MW: 360 kDa) and sodium hydroxide (NaOH, Purity: >97%) were procured from Sigma Aldrich.

### 2.2. Filtration Procedure

#### 2.2.1. PES UF Membrane

The filtration test of the PES UF membrane was conducted with three conventional cross-flow cells, each with an effective membrane area of 2.04 × 10^−3^ m^2^, connected in series, as illustrated in Figure 1. A detailed description of the membrane cells and the filtration system is presented in the work of Mosqueda-Jimenez et al. [19] and Zoka [20]. The feed flow rate of 1.1 L/min was maintained to minimize the concentration polarization effect [21].

The PES UF membrane filtration testing procedures were as follows:
(a)Measurement of PWF in the normal position: initially, the membrane coupons were pre-compacted for 1 h by filtering pure water at 70 psig (equivalent to 4.8 bar), followed by a pure water permeation test at 50 psig (3.4 bar) for 4 h. (b)Measurement of PWF in the inverted position: pre-compaction of new membrane coupons installed in the permeation cells under pure water pressure of 15 psig (1.0 bar) for 1 h, followed by a pure water permeation test at 5 psig (0.34 bar). For the permeation test, the permeate volume, *V* (L), collected during the period *t* (h) was measured, and the pure water flux PWF (L/m^2^·h) of the membrane was calculated by
(1)$$WF=\frac{V}{A\times t}$$
where *A* is the effective membrane area (2.04 × 10^−3^ m^2^). The average of the values obtained from the three permeation cells is reported.(c)UF filtration of a BSA solution in the normal position: pre-compaction of new membrane coupons was conducted with pure water at 70 psig for 1 h in the permeation cells, followed by the filtration experiment with a 100 mg/L BSA solution at 50 psig. (d)UF filtration of a BSA solution in the inverted position: pre-compaction of new membrane coupons in the permeation cells was conducted by filtering pure water at a pressure of 15 psig for 1 h; this was followed by the filtration experiment with a 100 mg/L BSA solution at 5 psig.(e)UF filtration of a BSA solution in the normal position: the coupons used in (b) were inverted back to the normal position. PWF was measured, and then a 100 mg/L BSA solution was filtered through them at a pressure of 50 psig. In the BSA filtration test, the permeate flux *J* was obtained by Equation (1), replacing PWF with *J*. The BSA concentration in the feed and permeate samples was measured using a UV spectrophotometer (DR6000, Hach Instruments, Loveland, CO, USA) at a wavelength of 220 nm. The solute separation, *R*, was then calculated by:(2)$$R=1-\frac{C_{p}}{C_{f}}$$
where *C_p_* and *C_f_* are permeate and feed solute concentrations, respectively.

#### 2.2.2. Ceramic UF Membrane

The filtration test of the ceramic membrane was conducted via a cross-flow filtration system using three different types of feed solutions (i.e., pure water, PVPK90, and BSA solution) to investigate the membrane water flux in both normal and inverted positions. The inner surface area of the tubular membrane (0.155 m^2^) was considered as the effective surface area, since the dense skin layer, which was brought into contact with the feed when the membrane was used in the normal position, was on the lumen side. Figure 2a,b illustrates the experimental setup for ceramic membrane filtration in normal and inverted positions, respectively.

The details of ceramic UF membrane filtration experiments are as follows:(a)Testing the ceramic UF membrane in the normal position: the tubular ceramic membrane was first installed and connected to the cross-flow system (membrane effective area: 0.155 m^2^). The feed water was first delivered to the lumen side of the membrane tubes at a pressure of 2 bar until a stable permeate flux was attained. This normally required 15 min. It was followed by reducing the pressure to 1 bar prior to the collection of the permeate for analysis. Then, the permeate was collected multiple times for a period of 30 min to calculate the permeate flux, and the average permeate flux was then reported.(b)Testing the ceramic UF membrane in the inverted position: the feed water flowed from the outer surface of the entire tubular ceramic membrane at 2 bar for 15 min in order to achieve flux stabilization. It was followed by reducing the pressure to 1 bar, and the permeate was collected multiple times for a period of 30 min to calculate the average permeate flux. The PWF of the membrane in both orientations was calculated based on Equation (1) as described for the PES UF membrane.(c)UF of a BSA solution in the normal position: the ceramic membrane (membrane effective area: 0.155 m^2^) was pre-treated using pure water at 2 bar for 15 min followed by the filtration of a 500 mg/L BSA solution at 1 bar. The permeate flux was recorded every 15 min for up to 60 min to yield an average permeate flux. The fouled membrane was then cleaned by filtering a 1 wt/v% NaOH solution for 30 min.(d)UF filtration of a BSA solution in the inverted position: the ceramic membrane (membrane effective area: 0.155 m^2^) was first pre-treated as in (b) at 2 bar for 15 min, followed by filtration of a 500 mg/L BSA solution at 1 bar. The permeate flux was recorded every 15 min for up to 60 min to yield an average permeate flux. The fouled membrane was then washed using a 1 wt/v% NaOH solution for 30 min. An alkaline solution was used because it is normally used to remove organic foulants from a membrane surface. The permeate flux *J* was calculated using Equation (1) by replacing PWF with *J*. The BSA concentration in the feed and permeate samples was measured using a UV–vis spectrophotometer (Nicolet iS10, Thermo Scientific, Voltam, MA, USA). The BSA rejection efficiency, *R*, was then calculated using Equation (2).(e)UF testing with the PVPK90 solution in normal position and inverted position: the protocol was similar to that for the filtration of a BSA solution, except that the feed solution was replaced by a 250 mg/L PVPK90 solution. The permeate flux *J* was calculated using Equation (1) by replacing PWF with *J*. The PVPK90 concentration in the feed and permeate samples was measured using a total organic carbon (TOC) analyzer (OCT-L, Shimadzu, Japan). The PVPK90 rejection efficiency, *R*, was then calculated using Equation (2).

### 2.3. Characterization by SEM

PES UF membrane: a scanning electron microscope (SEM, Vegall XMU, Tescan, Warrendale, PA, USA) was used to investigate the surface and cross-sectional morphology of the membranes. For the cross-sectional images, the membrane was soaked in liquid nitrogen, and the frozen membrane was cut into pieces with a sharp scissor. In order to increase electron conductivity, the samples were gold-sputtered to a thickness of 10 nm in a coater (Q150T, Quorum, Lewes, UK).

Ceramic UF membrane: the structural morphology of the membrane surfaces and the cross-section was examined using a tabletop SEM (TM3000, Hitachi, Tokyo, Japan). Since ceramic membranes do not break in liquid nitrogen, a small hammer was used to crack one corner of the tubular membrane to obtain small membrane samples for both surface and cross-section observation. The ceramic membrane samples were also gold-sputtered to a thickness of 10 nm (SC7620, Quorum, UK) before being subjected to SEM analysis.

## 3. Results

### 3.1. Characterization by SEM

The cross-sectional and top surface SEM images of the polymeric flat sheet membrane are shown in Figure 3. This membrane consisted of a top skin layer, a porous sublayer, and a backing material (Figure 3a). Figure 3b shows that the top skin layer was relatively smooth. In the cross-sectional images of the tubular ceramic membrane (Figure 4a,b), three layers are visible, i.e., an inner most and least porous layer with a thickness of about 20 μm, a middle layer with larger pores, with a thickness of 20–50 μm, and a highly porous outer layer. The outer surface (Figure 4d) was more porous than the inner surface (Figure 4c), in which few dark spots were noticeable. They could be defective pores.

### 3.2. Filtration Experiments

#### 3.2.1. PES UF Membrane

The results of PWF experiments with the PES membrane in the normal and inverted positions are shown in Figure 5. For the normal position, even though the membrane was compacted, the flux decreased from the initial 194 L/m^2^·h to the final 170 L/m^2^·h, with an average of 182 L/m^2^·h (permeance = 52.8 L/m^2^·h·bar). Note that the data were collected at the operating pressure of 50 psig.

For the PES UF membrane in the inverted position and at an operating pressure of 5 psig, the initial PWF increased from 967 L/m^2^·h at 0.17 h to 1643 L/m^2^·h at 1 h, but thereafter remained constant. Note that the data were collected at the operating pressure of 5 psig; thus, the permeance increase from the normal to the inverted position was nearly two orders of magnitude (i.e., from ~52.8 to an average value of 4630 L/m^2^·h·bar). The initial flux increase in Figure 5 was probably due to pore expansion that took place during the first hour.

The filtration of a BSA solution by the new PES UF membrane in the normal position was then carried out. The flux decreased from the initial 91.2 L/m^2^ h to 77.5 L/m^2^ h in 4 h of operation, with an average of 84.3 L/m^2^ h (i.e., 24.46 L/m^2^·h·bar) and a standard deviation of 3.5 L/m^2^ h (Figure not shown), which is less than half of the PWF shown in Figure 5. Meanwhile, the solute BSA separation was calculated to be 96%, which was reasonable, given that the molecular weight of BSA is 66 kDa [22], and the membrane’s nominal MWCO was 33 kDa.

Figure 6a shows the results of the filtration of the BSA solution by the new PES UF membrane in the inverted position. The flux increased initially as observed for the PWF (Figure 5), i.e., it increased from 1033 L/m^2^·h at 0.17 h to 1646 L/m^2^·h at 0.5 h, before gradually decreasing to 1296 L/m^2^·h (3760 L/m^2^·h·bar) in 4 h. Thus, the flux was slightly lower than the PWF. BSA separation meanwhile was as low as 8.3%.

The next experiment was the PWF measurement using the PES UF membrane in b) in the inverted position at 5 psig. The membrane was flipped back to the normal position, followed by the BSA filtration test at 50 psig. As the results in Figure 6b of BSA filtration in the normal position show, the initial flux at 0.17 h was 85 L/m^2^·h and decreased slightly to 74.1 L/m^2^·h after 2 h. The average flux was 79.4 L/m^2^·h (23.03 L/m^2^·h·bar), almost the same as that of the fresh PES UF membrane, i.e., 84.3 L/m^2^·h (24.46 L/m^2^·h·bar) in the normal position. Thus, the following trends were observed for the PES UF membrane.

A two-order-of-magnitude increase in pure water permeance took place when the PES UF membrane was inverted from the normal position.Solute (BSA) separation became very low (8.3%) when the membrane was in the inverted position.The initial UF performance was recovered when the membrane was flipped back to the normal position.

Therefore, it can be concluded that the pores that had expanded when the PWP experiment was performed in the inversed position contracted back to their original size when the membrane was flipped back to the normal position.

#### 3.2.2. Ceramic UF Membrane

The UF experimental results of the commercial ceramic membrane are summarized in Table 1. Two identical modules were used for these tests. Only one flux value is shown for each experiment, but it represents the average of multiple measurements made during an operating period of 30 min, unless otherwise recorded. It should also be noted that the flow direction of the tubular ceramic membrane was changed, differently from the flat sheet PES membrane, without flipping the membrane. In the normal position, the dense skin layer of the lumen side was in contact with the feed solution, while in the inverted position, the outer surface of the module was in contact with the feed solution.

From Table 1, it can be observed that

For Module 1 experiments, the PWF of the membrane became 1.91 times higher by changing the flow direction from the normal to the inverted position.In the normal position, a severe permeance decrease (82%) was observed when a PVPK90 solution was used as the feed, with solute separation of 52%. In the inverted position, the permeance decrease was much less (30%), and there was no solute separation.

It should be noted that the Module 2 experiments were conducted only in the normal position. Module 2 was somewhat different in pure water permeance from for Module 1, indicating a difference in performance between modules. The permeability of the ceramic membrane was not impacted as much (relative to the PWF) by the filtration of a 500 mg/L BSA solution as by the filtration of a 250 mg/L solution (for module 1), i.e., it was 15.5% versus 82%. Cleaning with a 1% NaOH solution was quite effective after BSA filtration in that it recovered 90% of pure water permeability. The treatment by a 1% NaOH solution (cleaning and backwashing) was not as effective after the filtration of the 250 mg/L PVPK90 solution, as it only recovered 77.8% of the starting pure water permeability (i.e., 406 versus 522 L/m^2^·h·bar). A comparison between BSA rejection by the ceramic membrane and the polymeric membrane in different positions is presented in Appendix A, together with their respective BSA concentrations in the feed and permeate samples (Appendix A).

## 4. Discussion

### Model Development

The objective of the model simulation was to examine if the increase in pure water permeance from 316.7 L/m^2^·h·bar for the normal orientation of a ceramic membrane to 626.3 L/m^2^·h·bar (1.91 times) for the inverted orientation could be explained. Note that these results were obtained from Module 1 of a solid ceramic membrane, whose pore geometry was considered the same at both positions, in contrast to the polymeric PES membrane which is more flexible. For this purpose, the model was developed based on Bernoulli’s law. 

As illustrated in Figure 7, it was assumed that the membrane could be divided into two segments. The top segment, which faced the feed solution in the normal position, contained smaller-size pores than the bottom segment. It was also assumed that the pores had a circular cross section and each top segment pore was connected to one bottom segment pore. Let the top pore of radius be $r_{t}$, connected to the bottom pore with radius $r_{b}$, as shown in Figure 7. 

The entrance of the top pore is open to the bulk of feed solution at a pressure of $p_{f}$, while a mass of water exits from the bottom pore exit and will eventually join the bulk of permeate. The pressure change along the depth direction of the pore can be written as
(3)$$p_{f}-p_{t}^{\prime }=h_{c}^{\prime }+\frac{\rho v_{t}^{2}}{2}$$
(4)$$p_{t}^{\prime }-p_{t}^{\prime \prime }=JR_{t}$$
(5)$$p_{t}^{\prime \prime }-p_{b}^{\prime }=\frac{\rho v_{b}^{2}}{2}-\frac{\rho v_{t}^{2}}{2}$$
(6)$$p_{b}^{\prime }-p_{b}^{\prime \prime }=JR_{b}$$
(7)$$p_{b}^{\prime \prime }-p_{p}=\frac{\rho v_{p}^{2}}{2}-\frac{\rho v_{b}^{2}}{2}$$
where $p_{t}^{\prime },\;p_{t}^{\prime \prime },\;p_{b}^{\prime },\;\mathrm{and}\;p_{b}^{\prime \prime }$ are pressures at the location indicated in Figure 5; *v_t_* and *v_b_* are the velocity of water in the top and bottom pore, respectively; *v_p_* is the velocity of water in the droplet coming out from the bottom pore exit. J is the single pore mass flow rate of water (kg/s pore), and $R_{t}\;\mathrm{and}\;R_{b}$ are the resistance against the flow of water in the top and bottom layer, respectively. Note that Bernoulli’s law was applied in Equations (3), (5) and (7). $h_{c}^{\prime }$ is the head loss due to the sudden contraction of the flow channel from a large feed reservoir to a pore, which can be given as
(8)$$h_{c}^{\prime }=\frac{k\rho v_{t}^{2}}{4}$$

In Equation (7), $v_{p}$ is difficult to evaluate but was assumed to be a fraction, *α*, of *v_b_*, since the water mass coming out from the bottom pore was larger than the bottom pore radius. Then,
(9)$$p_{b}^{\prime \prime }-p_{p}=\frac{\rho {\left(\alpha v_{b}\right)}^{2}}{2}-\frac{\rho v_{b}^{2}}{2}$$

The parameter *k* in Equation (8) is also unknown and depends on the flow pattern at the top layer pore entrance.

From (3) + (4) + (5) + (6) + (9) and combining Equation (8):(10)$$p_{f}-p_{p}=\frac{k\rho v_{t}^{2}}{4}+JR_{t}+JR_{b}+\frac{\rho {\left(\alpha v_{b}\right)}^{2}}{2}$$

Writing $\Delta p=p_{f}-p_{p}$

From Equation (10)
(11)$$J=\frac{\Delta p-\frac{k\rho v_{t}^{2}}{4}-\frac{\rho {\left(\alpha v_{b}\right)}^{2}}{2}}{R_{1}+R_{2}}$$

Applying Poiseuille’s law for the flow of water in the pore
(12)$$R_{t}=\frac{8\eta \delta_{t}}{\pi \rho r_{t}{}^{4}}$$
(13)$$R_{b}=\frac{8\eta \delta_{b}}{\pi \rho r_{b}{}^{4}}$$

Hence,
(14)$$J=\frac{\Delta p-\frac{k\rho v_{t}^{2}}{4}-\frac{\rho {\left(\alpha v_{b}\right)}^{2}}{2}}{\frac{8\eta \delta_{t}}{\pi \rho r_{t}{}^{4}}+\frac{8\eta \delta_{b}}{\pi \rho r_{b}{}^{4}}}$$

From the equation of continuity,
(15)$$\frac{v_{b}}{v_{t}}={\left(\frac{r_{t}}{r_{b}}\right)}^{2}$$
(16)$$J=\frac{\Delta p-\frac{k\rho v_{t}^{2}}{4}-\frac{\rho \alpha^{2}}{2}v_{t}^{2}{\left(\frac{r_{t}}{r_{b}}\right)}^{4}}{\frac{8\eta \delta_{t}}{\pi \rho r_{t}{}^{4}}+\frac{8\eta \delta_{b}}{\pi \rho r_{b}{}^{4}}}=\frac{\Delta p-\frac{k\rho v_{t}^{2}}{4}-\frac{\rho \alpha^{2}}{2}v_{t}^{2}{\left(\frac{r_{t}}{r_{b}}\right)}^{4}}{\frac{8\eta \delta_{b}}{\pi \rho r_{t}{}^{4}}\left\{\left(\frac{\delta_{t}}{\delta_{b}}\right)+{\left(\frac{r_{t}}{r_{b}}\right)}^{4}\right\}}$$

Based on Equation (16), the ratio of the fluxes $\frac{J_{l/s}}{J_{s/l}}$, where $J_{l/s}$ is the flux of Case 1—with the top pore larger than the bottom pore (inverted orientation)—and $J_{s/l}$ is the flux of Case 2—with the top pore smaller than the bottom pore (normal orientation)—is given by
(17)$$\frac{J_{l/s}}{J_{s/l}}=\frac{T2}{T1}\times \frac{-Q\left(S+R\right)+\sqrt{Q^{2}{\left(S+R\right)}^{2}+PT1\Delta pr_{l}{}^{4}}}{-Q\left(S+R\right)+\sqrt{Q^{2}{\left(S+R\right)}^{2}+PT2\Delta pr_{l}{}^{4}}}$$
where
(18)$$P=\frac{\rho }{{\left(\pi \rho \right)}^{2}}$$
(19)$$Q=\frac{8\eta \delta_{s}}{\pi \rho }$$
(20)$$R={\left(\frac{r_{l}}{r_{s}}\right)}^{4}$$
(21)$$S=\frac{\delta_{l}}{\delta_{s}}$$
(22)$$T1=k+2\alpha^{2}{\left(\frac{r_{l}}{r_{b}}\right)}^{4}$$
(23)$$T2=k{\left(\frac{r_{l}}{r_{s}}\right)}^{4}+2\alpha^{2}$$

The derivation of Equation (17) is given in detail in the Appendix A. Note that the experimentally obtained flux ratio, $\frac{J_{l/s}}{J_{s/l}}$, was 1.91. There may be a number of pore geometries that can result in the above ratio by simulation. One of such examples is at *k* = 100, *α* = 0.1, *r_l_* = 10 μm, *r_s_* = 1.95 μm, and *δ_l_* = *δ_s_* = 20 μm, suggesting the presence of large defective pores. The large *k* value suggests the dominance of the head loss caused by the sudden contraction of flow channel from the liquid reservoir to the pore with size in the micrometer range.

In summary, it was possible to explain the flux increase from the normal to the inverted position by the Bernoulli’s law, even when the pore geometry did not change. One possibility is that the radius of the larger pore was five times as large as that of the smaller pore and a severe head loss occurred due to the sudden contraction of the flow channel when water entered the pore.

## 5. Conclusions

The following conclusions can be drawn from this work.

(1)It was observed that the increase in PWF by flipping the membrane from its normal to an inverted position occurred not only for RO or NF membranes but also for UF membranes.(2)For the ceramic membrane, whose pore geometry was unchanged by flipping, the increase in PWF was nearly equal to twofold.(3)The above increase was explained by a mathematical model based on Bernoulli’s law, even though the model could not identify the pore geometry precisely.(4)For the polymeric PES membrane, the increase was as high as two orders of magnitude. This suggests other reasons for such a large increase, possibly the expansion of the size of the small pores.(5)The separation of BSA by the PES membrane was above 90% when the membrane was used in the normal position but became less than 10% when the membrane was used in the inverted position.

The separation of PVPK90 was as high as 52% when the ceramic membrane was used in the normal position but with severe flux decrease. There was no solute separation with much less flux decrease when the membrane was used in the inverted position.

## Figures and Tables

**Figure 1 membranes-12-00129-f001:**
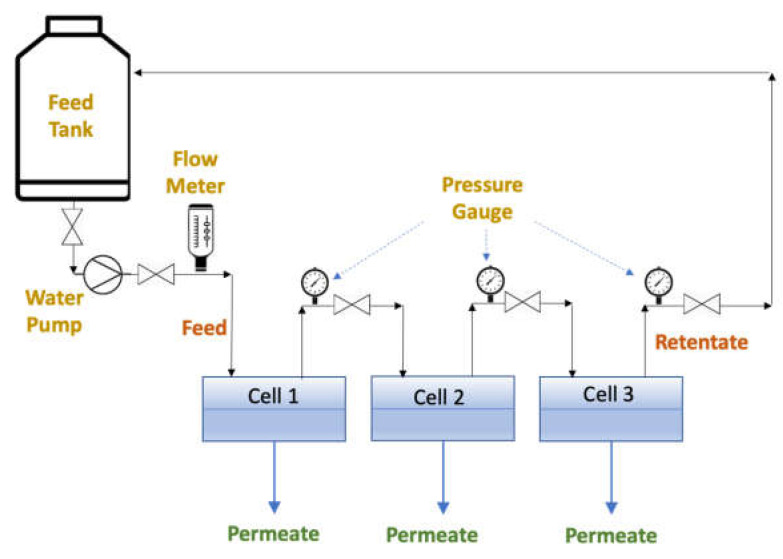
Filtration setup for flat sheet membranes with three permeation cells in series for both normal and inverted positions. In the normal position, the skin layer of the membrane faces upward in the cell, while it faces downward in the inverted position.

**Figure 2 membranes-12-00129-f002:**
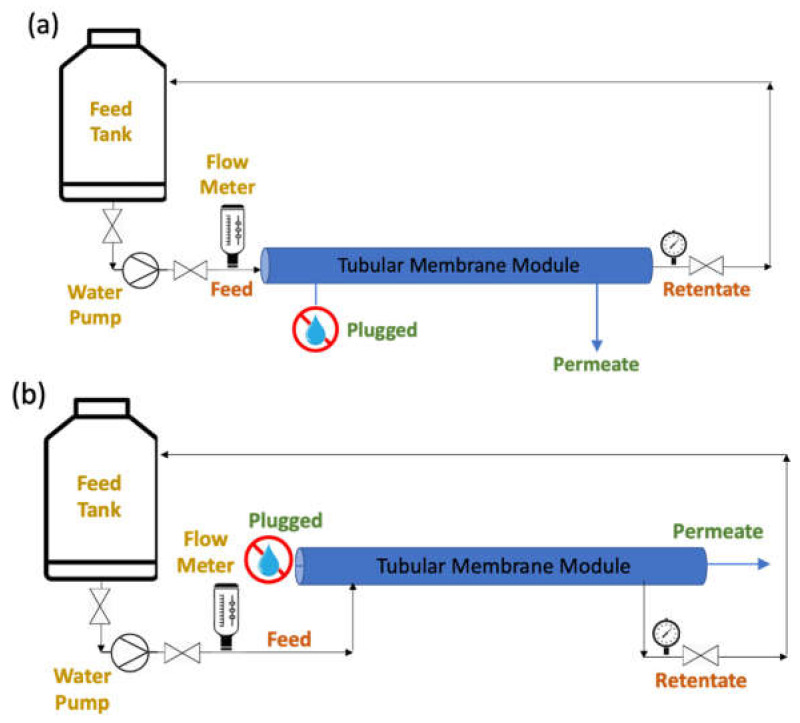
Filtration setup for tubular membranes, (**a**) normal position and (**b**) inverted position.

**Figure 3 membranes-12-00129-f003:**
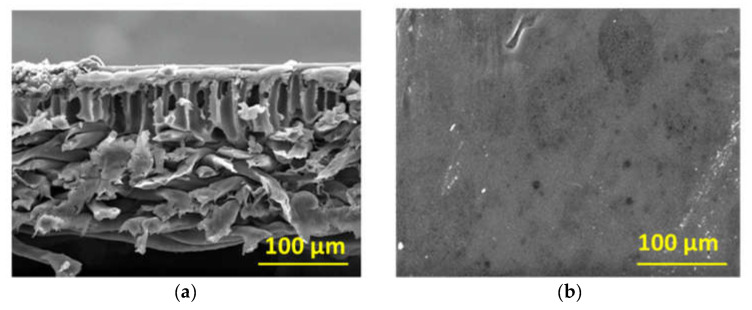
SEM images of (**a**) the cross-section and (**b**) the top surface of the PES UF membrane.

**Figure 4 membranes-12-00129-f004:**
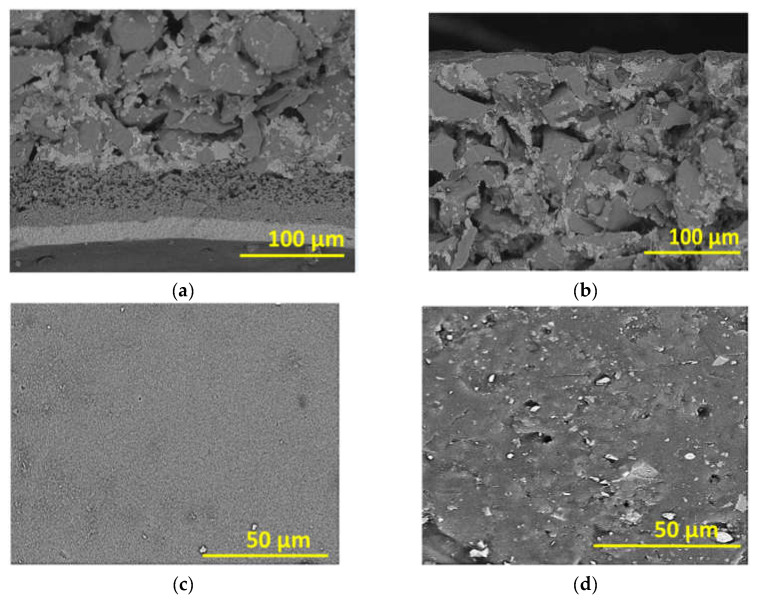
SEM images of the ceramic tubular membrane: (**a**) cross-section (adjacent to the inner surface), (**b**) cross-section (adjacent to the outer surface), (**c**) inner surface (selective), and (**d**) outer surfaces.

**Figure 5 membranes-12-00129-f005:**
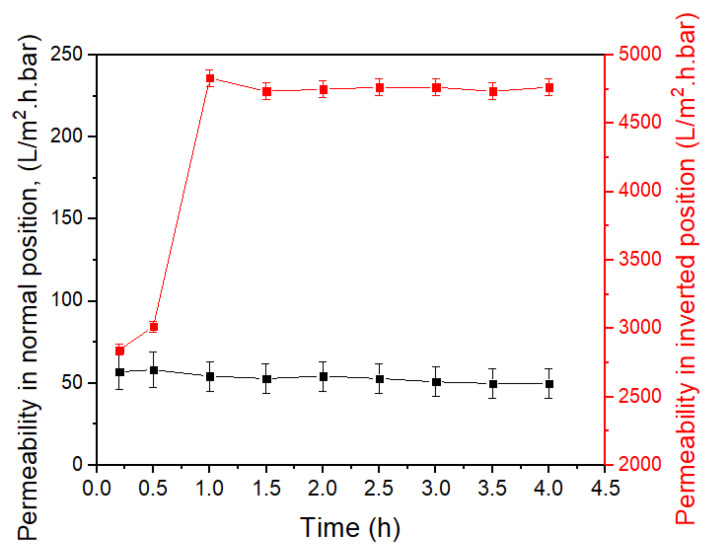
PWF of the PES UF membrane in the normal and inverted positions.

**Figure 6 membranes-12-00129-f006:**
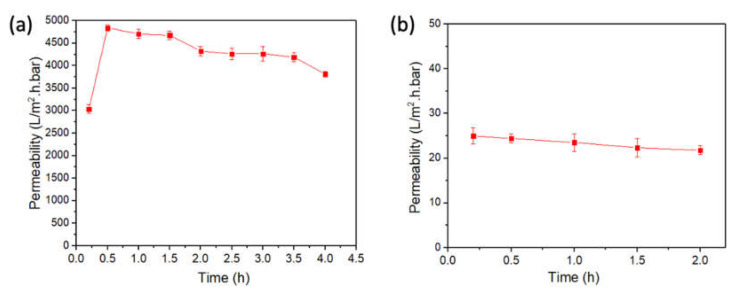
Flux change with time when a BSA solution (100 mg/L) was filtered by the UF membrane in the (**a**) inverted position (average of three coupons) and (**b**) normal position (the membrane was used for PWF measurement in the inverted position at 5 psig and then flipped back to the normal position).

**Figure 7 membranes-12-00129-f007:**
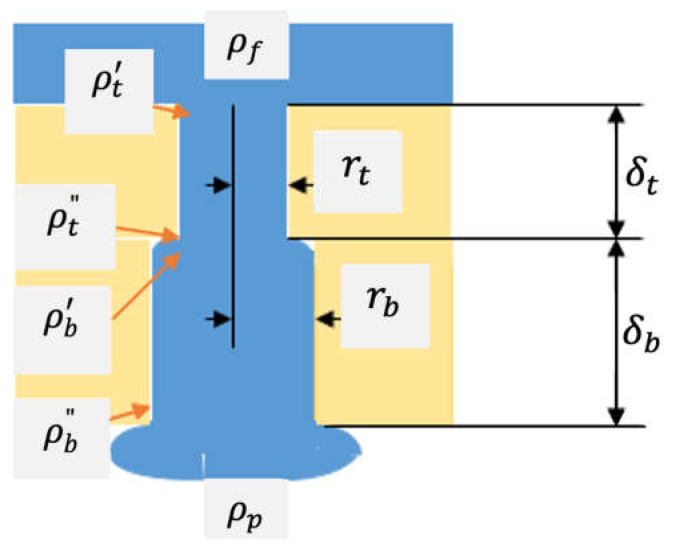
Schematic illustration of the porous structure where the top cylindrical pores (*t*) re connected to the bottom cylindrical pores (*b*) in series (only a top (small) pore and a bottom (large pore) are shown. The pores in the inverted position are not shown).

**Table 1 membranes-12-00129-t001:** Results of UF experiments with a commercial ceramic membrane run at 1 bar.

Position	Feed	Permeance ^a^ (L/m^2^·h·bar)	Separation (%)
** Module 1 **			
Normal position	Pure water	327.8 ± 4.5	-
Inverted position	Pure water	626.3 ± 10.9	-
Normal position	PVPK90 (250 ppm)	57.3 ± 7.6	52.0
Inverted position	PVPK90 (250 ppm)	437.6 ± 12.2	0
** Module 2 **			
Normal position	Pure water	580 ± 10.3	-
Normal position	BSA (500 ppm)	490 ± 19.5	40.0
Normal position after cleaning with 1 wt% NaOH solution	Pure water	522 ± 10.1	-
Normal position after cleaning with 1 wt% NaOH solution	PVPK90 (250 ppm)	-	52.0
Normal position after another cleaning with 1 wt% NaOH solution	Pure water	68.9 ± 1.1	-
Normal position after further backwashing with 1 wt% NaOH solution	Pure water	406 ± 3.0	-

^a^ Permeance (permeability) was calculated based on the inner surface area. Hereafter, the inner surface area is called the area.

## Data Availability

All the data are available inside this paper and in the Appendix A.

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
