# Peer review of "Flux Increase Occurring When an Ultrafiltration Membrane Is Flipped from a Normal to an Inverted Position—Experiments and Theory"

_membranes, 2022, doi:10.3390/membranes12020129_

Round 1

Reviewer 1 Report

The manuscript provides an interesting study on the investigation of “Flux Increase that Occurs when Ultrafiltration Membrane is Flipped from Normal to Inverted Position”, which will contribute to new understanding about the membrane flux. Some comments are listed below:

Major comments:

  • In the title “Flux Increase that Occurs when Ultrafiltration Membrane is Flipped from Normal to Inverted Position Experiments and Theory”. However, based on the results of current experiment, the PWF has increased to varying degrees of PES UF membrane and ceramic UF membrane. In this work, the mathematical model simulation can only explain the increase of the PWF of the ceramic ultrafiltration membrane, but cannot explain other types of ultrafiltration membranes. So, further systematic studies should be carried out to verify and explain the prevalence of this phenomenon in different UF membranes.
  • In the Abstract, the author should point out the purpose and perspectives of the present study to help to fast capture the potential readers interest.

[3] Generally, the membranes are not used in the inverted position as contacting its porous sublayer directly with

the feed solution could lead to severe fouling. The author continues to research on this basis, but lack of data directly proves the continuous improvement of this method in terms of membrane fouling, and it is difficult to show the practical significance of this research.

[4] A detailed description of the membrane cells and the filtration system the author should add in the supplementary materials. Please add the experimental setups.

[5] Figure 3 shows the PWF of the PES UF membrane in the normal position and the inverted position, but the two sets of data are performed under different operating pressures, the unit should be normalized and changed into

“L/m2.h.bar”. Please redraw the graph and conduct a comparative analysis.

[6] Data graphs related to the BSA separation should be supplemented and the BSA concentration in the feed and permeate samples should be displayed.

[7] Data related to“The filtration of BSA solution by the new PES UF membrane in the normal position” should be supplemented.

[8] In the Ceramic UF membrane section, the Module 2 experiments were conducted only in the normal position,considering the title , experiments should be added to consider the recovery of  PWF  in inverted position,and membrane pollution can be further analyzed.

[9] Please summarize directly in the discussion section how the mathematical model explained the increase in PWF from the normal to the inverted position.

[10] There is a lack of statistical analysis of the results.

[11] The Conclusions section was not well-organized. Just some main results were exhibited, which are short and not in-depth. Please reorganize it.

Minor comments:

[1] In the last paragraph of the Introduction section, a and b are listed, which are illogical, Please reorganized it.

[2] In the Materials section, “molecular weight cut off (MWCO)…”, please change“molecular weight cut off” to“molecular weight cutoff”.

[3] Page 3, Line 1-2: “The feed flow rate of 1.1 L/min was maintained so that the concentration polarization effect could be minimized”, Please add literature or other supporting materials.

[4] Page 3, Line 5: “52.8 to 4630 L/m2.h.bar”, Please add that “4630 L/m2.h.bar” is the average value of this set of data.

[5] Page 7, Line 3-4: The PWF data of the membrane was flipped back to the normal position should be supplemented.

Author Response

Reviewer 1

The manuscript provides an interesting study on the investigation of “Flux Increase that Occurs when Ultrafiltration Membrane is Flipped from Normal to Inverted Position”, which will contribute to new understanding about the membrane flux. Some comments are listed below:

Major comments:

  • In the title “Flux Increase that Occurs when Ultrafiltration Membrane is Flipped from Normal to Inverted Position Experiments and Theory”. However, based on the results of current experiment, the PWF has increased to varying degrees of PES UF membrane and ceramic UF membrane. In this work, the mathematical model simulation can only explain the increase of the PWF of the ceramic ultrafiltration membrane, but cannot explain other types of ultrafiltration membranes. So, further systematic studies should be carried out to verify and explain the prevalence of this phenomenon in different UF membranes.

Response: Thank you very much for your comment. It is believed that the flux increase at the inverted position occurs while maintaining the pore geometry in the case of ceramic membrane, while the pore expansion is taking place in the more flexible polymeric UF membrane. Since the mathematical model development is more complicated for the latter case, the discussion was limited to the ceramic membrane.

  • In the Abstract, the author should point out the purpose and perspectives of the present study to help to fast capture the potential readers interest.

Response: Thank you very much for your suggestion. We have modified the abstract accordingly by adding the following sentence to the abstract.

The flow from a layer of larger pore size to a layer of smaller pore size occurs in the backwashing of fouled membrane, and in forward and pressure retarded osmosis when the membrane is used in active layer facing the draw solution (AL-DS) mode. Therefore, this work is of practical significance for the cases where the direction of water flow is in the inverted position of membrane.”

[3] Generally, the membranes are not used in the inverted position as contacting its porous sublayer directly with the feed solution could lead to severe fouling. The author continues to research on this basis, but lack of data directly proves the continuous improvement of this method in terms of membrane fouling, and it is difficult to show the practical significance of this research.

Response: The practical significance of the research is shown in the revised abstract.

[4] A detailed description of the membrane cells and the filtration system the author should add in the supplementary materials. Please add the experimental setups.

Response: Thank you for your suggestion. The experimental setups for both polymeric and ceramic membranes are now presented in Figure 1 and 2 together with their respective description.

[5] Figure 3 shows the PWF of the PES UF membrane in the normal position and the inverted position, but the two sets of data are performed under different operating pressures, the unit should be normalized and changed into “L/m2.h.bar”. Please redraw the graph and conduct a comparative analysis.

Response: To be consistent, we have agreed to include standardized unit. The units of Figures 5 and 6 (all figures are renumbered in the revised version due to the addition of new figures) are now converted to ““L/m2.h.bar” and the standardized unit is also mentioned in text.

[6] Data graphs related to the BSA separation should be supplemented and the BSA concentration in the feed and permeate samples should be displayed.

Response: Agreed. A new figure comparing the BSA separation of polymeric and ceramic membrane in normal and inverted mode is now presented in Figure S1. The respective concentration of the BSA in feed and permeate samples are also presented in a table (Table S1).

[7] Data related to“The filtration of BSA solution by the new PES UF membrane in the normal position” should be supplemented.

Response: More detailed data for the filtration of BSA solution by PES UF membrane are given in the revised manuscript as follows.

The filtration of BSA solution by the new PES UF membrane in the normal position was then carried out. The flux decreased from the initial 91.2 L/m2 h to 77.5 L/m2 h in 4 h of operation with an average of 84.3 L/m2 h (i.e., 24.46 L/m2.h.bar) and a standard deviation of 3.5 L/m2 h (Figure not shown), which is less than half of the PWF shown in Fig. 3. Meanwhile, the solute BSA separation was calculated to be 96%.”

[8] In the Ceramic UF membrane section, the Module 2 experiments were conducted only in the normal position,considering the title , experiments should be added to consider the recovery of  PWF  in inverted position,and membrane pollution can be further analyzed.

Response: The most important experimental data from the ceramic UF membrane was that the PWF was nearly doubled from the normal to the inverted position of the Module although the pore geometry has not changed. The possibility of such phenomenon was further discussed in the theoretical section rigorously. We have added the data for the filtration of PVP and BSA solution where the flux decreased as expected. But these data are rather of secondary importance. Therefore, the experiments in the inverted position was not made with Module 2.

[9] Please summarize directly in the discussion section how the mathematical model explained the increase in PWF from the normal to the inverted position.

Response: Thank you very much for your comment. A brief summary was added to the theoretical section to explain how the increase in PWF occurs from the normal to the inverted position.

In summary, it is possible to explain the flux increase from the normal to the inverted position by Bernoulli’s law, even when the pore geometry does not change. One of such possibilities is that the radius of the larger pore is five times as large as the smaller pore and a severe head loss occurs due to the sudden contraction of the flow channel when water enters into the pore.”

[10] There is a lack of statistical analysis of the results.

Response: If the statistical analysis means the experimental error range, we have already shown the error bars to the flux data and also the range of the errors in Table 1. These data are obtained from different module (with length of 1 m)

[11] The Conclusions section was not well-organized. Just some main results were exhibited, which are short and not in-depth. Please reorganize it.

Response: Thank you very much for your suggestion. The conclusion was revised as follows.

  • It was observed that the increase in PWF by flipping the membrane from its normal to inverted position occurs not only for RO or NF membranes but also for UF membranes.
  • For the ceramic membrane where the pore geometry was unchanged by flipping the increase in PWF was nearly equal to twofold.
  • The above increase was explained by a mathematical model based on Bernoulli’s law, even though the model could not identify the pore geometry precisely.
  • For the polymeric PES membrane, the increase was as high as two orders of magnitude. This suggests that there is some other reason for such a large increase, possibly the expansion of the size of the small pore.
  • The separation of BSA by the PES membrane was above 90% when the membrane was used in the normal position but it became less than 10% when the membrane was used in the inverted position.
  • The separation of PVPK90 was as high as 52% when the ceramic membrane was used in the normal position but with severe flux decrease. There was no solute separation with much less flux decrease when the membrane was used in the inverted position.

Minor comments:

[1] In the last paragraph of the Introduction section, a and b are listed, which are illogical, Please reorganized it.

Response:  Thank you very much for your comment. According to the suggestion, the last paragraph of the introduction was rephrased as follows:

For this purpose, a) two flat sheet commercial polyethersulfone (PES) UF membranes and b) two tubular ceramic membrane modules were tested in their normal and inverted positions. For both polymeric and ceramic membranes filtration experiments were conducted to measure the pure water flux (PWF) and the flux of macromolecular solutions.”

[2] In the Materials section, “molecular weight cut off (MWCO)…”, please change“molecular weight cut off” to“molecular weight cutoff”.

Response: We have corrected the typo.

[3] Page 3, Line 1-2: “The feed flow rate of 1.1 L/min was maintained so that the concentration polarization effect could be minimized”, Please add literature or other supporting materials.

Response:  Thank you for your comment. We have included a reference to support statement. In the work conducted by Wei (Int. J. Mol. Sci. 11(2) (2010) 672-690), flow rates between 1 and 1.8 L/min were employed and our flow rate fell within the range.

[4] Page 3, Line 5: “52.8 to 4630 L/m2.h.bar”, Please add that “4630 L/m2.h.bar” is the average value of this set of data.

Response:  Agreed. We have mentioned that “4630 L/m2.h.bar” is the average value of the data set in the revised manuscript.

[5] Page 7, Line 3-4: The PWF data of the membrane was flipped back to the normal position should be supplemented

Response:  Thank you for your comment. The result is now provided, i.e., the average flux was 79.4 L/m2.h (23.03 L/m2.h.bar), which is almost the same as that of the fresh PES UF membrane, i.e., 84.3 L/m2.h (24.46 L/m2.h.bar) in the normal position.

Reviewer 2 Report

Review of the article entitled Flux Increase that Occurs when Ultrafiltration Membrane is Flipped from Normal to Inverted Position - Experiments and Theory

Authors: Ladan Zoka, Ying Siew Khoo, Woei Jye Lau, Takeshi Matsuura,
Roberto Narbaitz, Ahmad Fauzi Ismail

The manuscript is a continuation of the research, in which the Authors focused on the impacts of flipping the membranes of dense/porous layer asymmetry. Two types of membranes (ceramic vs. polymeric) were investigated, especially in terms of pure water flux and separation properties. The Authors employed the Bernoulli’s law to explain the increasing of pure water flux in the inverted position. The paper is written in a concise form. I find the paper suitable for publication in Membranes journal. However, I recommend the manuscript to be published after major revision – it should be carefully revised according to the specific comments presented below:

  1. For the convenience of the revision the line numbers should be added.
  2. In the Introduction section the Authors stated that the effects of flipping the membranes with hydrophilic/hydrophobic asymmetry are well documented in the literature. Therefore, the Authors should unequivocally mark the aim of their work and emphasize the novelty of their investigations. Now this issue remains unclear. The Authors should present the state of art in this subject and, on this background, show what distinguishes their study from the previous ones. From this point of view the presented literature review is too general and not sufficient.
  3. Page 2, Introduction – Please explain the meaning of the arrows.
  4. 2. Materials - Please provide the material of ceramic membrane.
  5. 2.1. and 2.2. - Why did the Authors choose and compare the membranes with such difference in effective surface area?
  6. The resolution of the SEM microphotographs should be provided.
  7. The whole manuscript seems like a lab report, containing only the description of experimental results without deep discussion. Please enhance the discussion section with proper literature.
  8. The resolution of figures should be improved.
  9. The Conclusions section should be more comprehensive.

Author Response

The manuscript is a continuation of the research, in which the Authors focused on the impacts of flipping the membranes of dense/porous layer asymmetry. Two types of membranes (ceramic vs. polymeric) were investigated, especially in terms of pure water flux and separation properties. The Authors employed the Bernoulli’s law to explain the increasing of pure water flux in the inverted position. The paper is written in a concise form. I find the paper suitable for publication in Membranes journal. However, I recommend the manuscript to be published after major revision – it should be carefully revised according to the specific comments presented below:

1. For the convenience of the revision the line numbers should be added.

Response: Line numbers are now included in the revised manuscript.

2. In the Introduction section the Authors stated that the effects of flipping the membranes with hydrophilic/hydrophobic asymmetry are well documented in the literature. Therefore, the Authors should unequivocally mark the aim of their work and emphasize the novelty of their investigations. Now this issue remains unclear. The Authors should present the state of art in this subject and, on this background, show what distinguishes their study from the previous ones. From this point of view the presented literature review is too general and not sufficient.

Response: Thank you for your comments. We have now improved the content of Introduction to emphasize the novelty of our investigation.

3. Page 2, Introduction – Please explain the meaning of the arrows.

Response: We have revised the statement to avoid confusing.

4. Materials - Please provide the material of ceramic membrane.

Response: The materials of ceramic membrane is now provided.

5. 2.1. and 2.2. - Why did the Authors choose and compare the membranes with such difference in effective surface area?

Response: Thank you for your comment. The effective surface area of ceramic membrane is larger compared to the polymeric membrane as we could not purchase small size of ceramic membrane from the market. Since the flux of both types of membranes is represented by the same unit (LMH/bar), it is generally acceptable.

6. The resolution of the SEM microphotographs should be provided.

Response: The scale bar (either 100 or 50 micron) could be found in the respective images in Figure 3 and 4.

7. The whole manuscript seems like a lab report, containing only the description of experimental results without deep discussion. Please enhance the discussion section with proper literature.

Response: From the title, our work is about the experiments and theory behind the flux change when membrane is flipped from its normal position to inverted position. Some parts of the discussion are now improved according to Reviewer 1 and 3’s inputs.

8. The resolution of figures should be improved.

Response: Image resolution of Figure 7 is now improved.

9. The Conclusions section should be more comprehensive

Response: We have improved it accordingly.

Reviewer 3 Report

General comments:

This manuscript aims to show that increase in pure water flux by flipping the membrane from its normal to inverted position may occur not only for RO or NF membranes (as shown in previous studies) but also for UF membranes. This phenomenon is demonstrated with two commercial membranes, i.e. polymeric flat-sheet membrane (with MWCO of 30 kDa) and ceramic tubular  membrane (with MWCO of 100 kDa). Filtration experiments with model compound solutions showed how flipping altered filtration behavior of the two studied membranes. Mathematical model based on the Bernoulli’s law is also proposed to explain why PWF may increase when membrane is flipped. 

The manuscript is clear, topic is quite interesting and information is presented in a well-structured manner. Introduction is well-written. The experimental design has been appropriate to respond to research aims and results have been reproducible. Weaknesses of this manuscript are lack of originality of the research aims and interpretation of the results which could posses more scientific depth. Conclusions of the manuscript are consistent with the research aims and results. 

Comments regarding research aims and execution of this study:

The first research aim (proving that flipping of the UF membranes can lead to increase in flux) is experimented with two (more or less randomly selected?)  commercial membranes which have nothing in common but the MWCO in the range of UF membranes. Information regarding the support layer material is missing in both cases, as well as information regarding the skin layer material of the polymeric membrane. If the material of support layer and skin layer were the same it should be pointed out. It would be also interesting to know why these two membranes where chosen and why e.g. polymeric and ceramic membranes with the same MWCO were not selected (which would have enabled some comparison if the filtration pressures would have been about the same)?

The second research aim (to show how two different membranes behave in filtration conditions when membrane is used normally or flipped) is studied with the support of two different cases. PWF measurements show interestingly that in both cases flipping of the membrane causes significant increase in flux. It remains unclear whether this is atypical behavior for the UF membranes or whether this tends to happen always. Therefore, expanding the data set with some other UF membranes might provide valuable information regarding the commonness of the studied phenomenon. Filtration results of BSA and PVP solutions are not comparable between the two cases because concentration of BSA was different for the two different membranes. Moreover, ceramic membrane was chemically cleaned after the filtration with BSA and experiments with PVP were conducted only with ceramic membrane. Could authors please explain why  such different experimental conditions were chosen for the experiments with model compounds? 

The third research aim (to show mathematically that it is theoretically possible that certain pore geometries might result increase in flux when membrane is flipped) is very interesting but unfortunately, the model is not really validated with experimental work. It would be nice to see if experimental results with certain membrane with filtration conditions and certain parameters (the ones included in your model) would fit with the results of the model. Could you please comment if it would be practically possible to validate your model?

Other comments:

1. Introduction

Introduction is clear and well-written.

2 Materials and Methods

2.1 Materials

Was the 30 kDa supplied UF membrane supplied by Synder Filtration so called MK membrane? (detailed information regarding the membrane "name" is missing). 

What was the material of "ceramic tubular UF membrane with a nominal MWCO of 100 kDa supplied by Shandong Jinhuimo Technology Co. Ltd." and does the membrane have some "name" or catalogue number etc. that could help to identify it? 

Polyvinylpyrrolidone (PVPK90) was used as filtration model compound in the experiments conducted with ceramic membrane. PVP is not that commonly used as model compound and thus, I'm curious to hear why it was chosen and used only with ceramic membrane (and not with PES membrane)?  

Subsection 2.2.1. PES UF membrane:

In the beginning of subsection 2.2.1 (page 3) it is really nice that authors have shown the pressure values used both as psig and bar values. This helps a lot readers more familiar with bar units. Unfortunately, this is discontinued after a) and b) parts of the description of the filtration procedure. Therefore, I'm requesting authors to keep a consistent (and reader-friendly) style throughout the manuscript and to include bar values everywhere into brackets after psig values. 

Points c), d) and e) For how long the UF filtration of 100 mg/L BSA solution was conducted in each case (in hours)?

It is mentioned in the beginning of subsection 2.2.1 that "the feed flow rate of 1.1 L/min was maintained so that the concentration polarization effect could be minimized." However, the pressures used in the filtration of BSA solution in the normal position and in the inverted position had 10 fold difference, 50 psig versus 5 psig respectively. How do authors see the effect of this considerable pressure difference on the concentration polarization and fouling caused by BSA? 

Subsection 2.2.2. Ceramic UF membrane:

Points c), d) and e) For how long the UF filtration of 500 mg/L BSA solution or 250 mg/L of PVPK90 solution was conducted in each case (in hours)?

Why the used BSA concentrations were different for PES membrane and ceramic membrane? (100 mg/L and 500 mg/L). 

In point c) it is mentioned that after normal position filtration of BSA solution the fouled ceramic membrane was cleaned by filtering a 1 wt/v% of NaOH solution for 30 min. Why membrane was cleaned in this case?

3. Results

Figures and table are appropriate and support text but captions could be a bit more detailed. For example names and MWCO's of the membranes could be mentioned and in the case of flux figures information regarding used filtration conditions would make figures more self-sufficient and optimized for image searches. It should be checked whether resolution of flux graphs and Figure 5 meet the criteria of this journal (because now its much less than commonly required 400 dpi). Also, symbols presented in Figure 5 should be opened in the caption.

4. Discussion

4.1. Model development

In the discussion chapter filtration model is being developed but not really validated which is unfortunate. The developed model shows that the ratio of the fluxes inverted and normal position can be somehow simulated. However, comparison of the experimental result (i.e. measured flux ratio of 1.91) to the model does not really validate the model but only shows that such ratio is also theoretically possible to be acquired if combinations of input parameters happen to be suitable, i.e. "There may be a number of pore geometries that can result in the above ratio by simulation" as authors have stated. 

5. Conclusions

Mostly aligned with research aims and results. Besides changes in PWF values effect of flipping on rejection of membrane could be also highlighted both here and in abstract if there no limiting space restrictions. 

References

Membranes journal asks reviewers to look out for inappropriate (=excessive?) self-citations. In this case 7 of the 23 citations (=30 %) cite previous publications of the authors of this paper which is rather high share. Citations [3], [19], [21] and [23] seem to be perfectly eligible but could authors please justify inclusion of references [5], [10] and [11]? Do those clearly improve the quality of this manuscript?

Author Response

General comments:

This manuscript aims to show that increase in pure water flux by flipping the membrane from its normal to inverted position may occur not only for RO or NF membranes (as shown in previous studies) but also for UF membranes. This phenomenon is demonstrated with two commercial membranes, i.e. polymeric flat-sheet membrane (with MWCO of 30 kDa) and ceramic tubular  membrane (with MWCO of 100 kDa). Filtration experiments with model compound solutions showed how flipping altered filtration behavior of the two studied membranes. Mathematical model based on the Bernoulli’s law is also proposed to explain why PWF may increase when membrane is flipped. 

The manuscript is clear, topic is quite interesting and information is presented in a well-structured manner. Introduction is well-written. The experimental design has been appropriate to respond to research aims and results have been reproducible. Weaknesses of this manuscript are lack of originality of the research aims and interpretation of the results which could posses more scientific depth. Conclusions of the manuscript are consistent with the research aims and results.

Response:  Thank you for your comments with constructive criticisms. 

Comments regarding research aims and execution of this study:

The first research aim (proving that flipping of the UF membranes can lead to increase in flux) is experimented with two (more or less randomly selected?)  commercial membranes which have nothing in common but the MWCO in the range of UF membranes. Information regarding the support layer material is missing in both cases, as well as information regarding the skin layer material of the polymeric membrane. If the material of support layer and skin layer were the same it should be pointed out. It would be also interesting to know why these two membranes where chosen and why e.g. polymeric and ceramic membranes with the same MWCO were not selected (which would have enabled some comparison if the filtration pressures would have been about the same)?

Response:  Regarding the polymeric membrane both skin and support layer were made of PES. The skin and support layer of the ceramic membrane were made of alumina and zirconia. Unfortunately, ceramic membrane of the same molecular weight cut-off as that of the PES membrane was not available.

The second research aim (to show how two different membranes behave in filtration conditions when membrane is used normally or flipped) is studied with the support of two different cases. PWF measurements show interestingly that in both cases flipping of the membrane causes significant increase in flux. It remains unclear whether this is atypical behavior for the UF membranes or whether this tends to happen always. Therefore, expanding the data set with some other UF membranes might provide valuable information regarding the commonness of the studied phenomenon. Filtration results of BSA and PVP solutions are not comparable between the two cases because concentration of BSA was different for the two different membranes. Moreover, ceramic membrane was chemically cleaned after the filtration with BSA and experiments with PVP were conducted only with ceramic membrane. Could authors please explain why  such different experimental conditions were chosen for the experiments with model compounds? 

Response:  Thank you very much for the insightful comments. The flux increase by flipping the membrane seems very common as we have already reported the similar phenomena for RO and NF membranes. It would definitely be interesting to do similar investigations for UF membrane of different MWCOs.

BSA experiments are common for both polymeric and ceramic membranes. PVP of higher molecular weight was chosen for the filtration test of the ceramic membrane because of the higher MWCO of the ceramic membrane.

The third research aim (to show mathematically that it is theoretically possible that certain pore geometries might result increase in flux when membrane is flipped) is very interesting but unfortunately, the model is not really validated with experimental work. It would be nice to see if experimental results with certain membrane with filtration conditions and certain parameters (the ones included in your model) would fit with the results of the model. Could you please comment if it would be practically possible to validate your model?

Response:  This question is replied in response to the question asked below regarding section 4.1. on model development.

Other comments:

  1. Introduction

Introduction is clear and well-written.

Response: Thank you very much for your comment.

2 Materials and Methods

2.1 Materials

Was the 30 kDa supplied UF membrane supplied by Synder Filtration so called MK membrane? (detailed information regarding the membrane "name" is missing).

Response:  Yes, it is MK membrane. The information was added in Materials. 

What was the material of "ceramic tubular UF membrane with a nominal MWCO of 100 kDa supplied by Shandong Jinhuimo Technology Co. Ltd." and does the membrane have some "name" or catalogue number etc. that could help to identify it? 

Response:  Thank you for your comments. The model of the ceramic membrane we purchased from the company is T7/60/250. This information is now added into the revised MS. Besides, the materials (i.e., alumina and zirconia) used to fabricate the ceramic membrane are also provided in the main text.

Polyvinylpyrrolidone (PVPK90) was used as filtration model compound in the experiments conducted with ceramic membrane. PVP is not that commonly used as model compound and thus, I'm curious to hear why it was chosen and used only with ceramic membrane (and not with PES membrane)?  

Response:  Thank you for your comment. It is correct that PVKK90 is not commonly used as model compound, but in the case of ceramic membrane which has significantly larger MWCO (100 kDa) compared to the PES membrane (30 kDa), using PVPK90 could provide better insights toward the membrane separation rate. Because of this reason, for ceramic membranes we tested two different solutes (i.e., BSA and PVPK90) instead of single type of solute (BSA) as in PES membrane.

Subsection 2.2.1. PES UF membrane:

In the beginning of subsection 2.2.1 (page 3) it is really nice that authors have shown the pressure values used both as psig and bar values. This helps a lot readers more familiar with bar units. Unfortunately, this is discontinued after a) and b) parts of the description of the filtration procedure. Therefore, I'm requesting authors to keep a consistent (and reader-friendly) style throughout the manuscript and to include bar values everywhere into brackets after psig values. 

Response:  Thank you for your comment. To be consistent, we converted the unit to “LMH/bar”. The changes can be found in Figure 5 and 6. In addition, we have included “LMH/bar” in the main text as well.

Points c), d) and e) For how long the UF filtration of 100 mg/L BSA solution was conducted in each case (in hours)?

It is mentioned in the beginning of subsection 2.2.1 that "the feed flow rate of 1.1 L/min was maintained so that the concentration polarization effect could be minimized." However, the pressures used in the filtration of BSA solution in the normal position and in the inverted position had 10 fold difference, 50 psig versus 5 psig respectively. How do authors see the effect of this considerable pressure difference on the concentration polarization and fouling caused by BSA? 

Response:  Concentration polarization is usually expressed in terms of exp(v/k), where k is the mass transfer coefficient and v is the permeation velocity. k largely depends on the feed flow rate and for this system it is known to be in the range of 10-4 m/s. The permeation velocity for the normal position is about 80 kg/m2 h, which is nearly equal to 2 x10-5 m/s and an order of magnitude smaller than k. The concentration polarization is low but cannot be ignored. Therefore, we said that the concentration polarization effect was minimized. In the case of the inverted position, the separation was so low that the concentration polarization can be safely ignored.

Subsection 2.2.2. Ceramic UF membrane:

Points c), d) and e) For how long the UF filtration of 500 mg/L BSA solution or 250 mg/L of PVPK90 solution was conducted in each case (in hours)?

Response:  Thank you for your comment. Depending on the cases, the data obtained were either average of 30-min  filtration or 60-min filtration. This information is now highlighted in the text.

Why the used BSA concentrations were different for PES membrane and ceramic membrane? (100 mg/L and 500 mg/L). 

Response:  Thank you for pointing it out. As the effective surface area of the ceramic membrane (0.155 m2) is significantly larger compared to the PES membrane (0.00204 m2), we needed to use higher BSA concentration to obtain an accurate data. BSA is likely to be adsorbed onto the ceramic membrane and if low concentration is used, we could not interpret correctly the rejection of the membrane.

In point c) it is mentioned that after normal position filtration of BSA solution the fouled ceramic membrane was cleaned by filtering a 1 wt/v% of NaOH solution for 30 min. Why membrane was cleaned in this case?

Response:  Thank you for your comment. Alkaline solution is normally used to remove organic foulants from the membrane surface and in this case, we only used low concentrated NaOH solution as cleaning agent as our membrane filtration experiment was only conducted for short duration. Such cleaning approach is found to be effective as membrane water flux could be highly retrieved upon NaOH cleaning (see Table 1). We have included the reason why alkali solution was used in this study in Section 2.2.2.

  1. Results

Figures and table are appropriate and support text but captions could be a bit more detailed. For example names and MWCO's of the membranes could be mentioned and in the case of flux figures information regarding used filtration conditions would make figures more self-sufficient and optimized for image searches. It should be checked whether resolution of flux graphs and Figure 5 meet the criteria of this journal (because now its much less than commonly required 400 dpi). Also, symbols presented in Figure 5 should be opened in the caption.

Response:  Thank you for your comments. We have now improved the caption of the figures. Besides, we have improved the image resolution of Figure 5 (now Figure 7 in the revised version) so as it can meet the scientific standard.

  1. Discussion

4.1. Model development

In the discussion chapter filtration model is being developed but not really validated which is unfortunate. The developed model shows that the ratio of the fluxes inverted and normal position can be somehow simulated. However, comparison of the experimental result (i.e. measured flux ratio of 1.91) to the model does not really validate the model but only shows that such ratio is also theoretically possible to be acquired if combinations of input parameters happen to be suitable, i.e. "There may be a number of pore geometries that can result in the above ratio by simulation" as authors have stated. 

Response:  Thank you very much for the reviewer’s insightful comment. Yes, we have to admit that the model can only suggest one of the possible pore geometries and cannot pinpoint the geometry. Nevertheless, the model shows that it is not impossible to explain the experimentally observed phenomenon by the Bernoulli’s law.

  1. Conclusions

Mostly aligned with research aims and results. Besides changes in PWF values effect of flipping on rejection of membrane could be also highlighted both here and in abstract if there no limiting space restrictions.

Response:  Thank you very much for your comment. The reviewer 1 also suggested the change of conclusion. Please look at our response to reviewer 1’s comment.  

References

Membranes journal asks reviewers to look out for inappropriate (=excessive?) self-citations. In this case 7 of the 23 citations (=30 %) cite previous publications of the authors of this paper which is rather high share. Citations [3], [19], [21] and [23] seem to be perfectly eligible but could authors please justify inclusion of references [5], [10] and [11]? Do those clearly improve the quality of this manuscript?

Response:  Thank you for your concerns. We have decided to remove Ref [5], [10] and [11] from the main text to improve the quality of the paper scientifically.

Round 2

Reviewer 1 Report

Accept 

Reviewer 2 Report

The revised manuscript is acceptable.

Reviewer 3 Report

Thank you for your replies and comments. After the revisions paper is acceptable. However, I still tend to think that discussion of results could possess more scientific depth and proves regarding generality of the studied phenomena in the case of UF membranes would strengthen this work. 

My final last but not least comment is: Please check once more whether the figures meet the resolution criteria of 300 dpi set up in the Instructions for Authors. It seems that also another reviewer pointed out also that resolutions of figures could be better and e.g. flux figures look still ugly pixelated.